# Baseline Analysis of Serotype-Specific IgG Antibody Levels for 13-Valent Pneumococcal Conjugate Vaccine in Healthy Chinese Individuals: A Multicenter Retrospective Study

**DOI:** 10.3390/vaccines13080847

**Published:** 2025-08-10

**Authors:** Gang Shi, Hong Li, Lina Guo, Lin Yuan, Jingjing Chen, Bin Li, Jinbo Gou, Weiyan Yin, Shuquan Luo, Jing Ti, Mengqi Duan, Fang Cao, Xiao Xu, Bin Wang

**Affiliations:** 1National Institutes for Food and Drug Control, NHC Key Laboratory of Research on Quality and Standardization of Biotech Products, State Key Laboratory of Drug Regulatory Science, NMPA Key Laboratory for Quality Research and Evaluation of Biological Products, Beijing 102629, China; shigang@nifdc.org.cn (G.S.); lihong@nifdc.org.cn (H.L.); guolina@nifdc.org.cn (L.G.); libin@nifdc.org.cn (B.L.); 2Walvax Biotechnology Co., Ltd., Kunming 650106, China; ynwsyl@walvax.com (L.Y.); ynwscjj@walvax.com (J.C.); yinweiyan@walvax.com (W.Y.); 3CanSino Biologics Inc., Tianjin 300457, China; jinbo.gou@cansinotech.com; 4Lanzhou Institute of Biological Products Co., Ltd., Lanzhou 730046, China; luoshuquan@sinopharm.com; 5Beijing Minhai Biotechnology Co., Ltd., Beijing 102600, China; tijing@biominhai.com; 6Fosun Adgenvax (Chengdu) Biopharmaceutical Co., Ltd., Chengdu 610219, China; duanmengqi@fosunpharma.com; 7AIM Vaccine Co., Ltd., Shanghai 201109, China; fang.cao@aimmbio.com

**Keywords:** streptococcus pneumoniae, capsule polysaccharide, ELISA, vaccine, baseline antibody level

## Abstract

Background/Objectives: The immunogenicity of Streptococcus pneumoniae vaccines is commonly evaluated by assessing the fold increase or proportions exceeding 0.35 μg/mL in serotype-specific IgG antibody levels post-vaccination. Establishing baseline antibody levels in unvaccinated populations is therefore essential for defining serological thresholds and understanding naturally acquired immunity. This study aimed to assess the seroprevalence and baseline levels of IgG antibodies specific to 13 pneumococcal capsular polysaccharide serotypes in healthy infants and young children across multiple regions of China from 2016 to 2023, supporting evidence-based PCV13 vaccination strategies. Methods: IgG concentrations for 13 serotypes (1, 3, 4, 5, 6A, 6B, 7F, 9V, 14, 18C, 19A, 19F, and 23F) were measured in unvaccinated individuals using the WHO-recommended ELISA. Univariate and multivariate analyses were applied to evaluate regional, age, and gender effects on baseline antibody levels. Results: GMCs for serotypes 6B, 14, 19A, and 19F exceeded 0.35 μg/mL, with 14 being the highest (1.64 μg/mL) and serotypes 3 and 4 the lowest. Significant regional variation (*p* < 0.001) and a U-shaped age trend were observed, with the lowest being at 7–11 months (*p* = 0.003). Conclusions: Baseline IgG levels varied by region and age. No significant gender differences were observed, and overall antibody levels were higher in the southern region.

## 1. Introduction

*Streptococcus pneumoniae* (Spn), commonly known as pneumococcus, is a Gram-positive bacterium that colonizes the mucosal surfaces of the upper respiratory tract in humans [1]. It is a leading cause of both invasive and non-invasive diseases, including pneumonia, otitis media, meningitis, and septicemia [2]. In China, *S*. *pneumoniae* is a leading pathogen of community-acquired pneumonia (CAP) and is the sole pathogen of invasive pneumococcal disease (IPD) in children, where IPD represents a specific diagnosis [3]. The capsular polysaccharide (CPS), located outside the bacterial cell wall, is the primary virulence factor of *S. pneumoniae*, contributing substantially to its pathogenicity [4,5]. CPS inhibits complement-mediated opsonization (C3b) and impairs phagocytic clearance by immune cells. Furthermore, CPS forms the basis for pneumococcal serotyping [6,7].

To date, over 100 pneumococcal capsular serotypes have been identified. The disease burden of pneumococcal infection exhibits marked geographic variation, with significantly higher under-five mortality rates in low- and middle-income countries [8]. China ranks among the top ten countries worldwide in terms of pneumococcal disease burden in children. WAHL et al. [9] estimated that in 2017, there were approximately 218,200 severe pneumococcal infections among Chinese children under five, including 8000 fatalities. Notably, mortality was disproportionately higher in western China, accounting for 49% of deaths [10]. According to national acute respiratory infection surveillance data from 2009 to 2019 [11], Spn was detected in 29.9% (8000/26,757) of patients testing positive for bacterial pathogens, with a detection rate of 38.5% (4049/10,517) among children, particularly those under five years of age.

Although antibiotics have long been the mainstay for treating pneumococcal infections, the emergence of antibiotic-resistant and multidrug-resistant Spn strains has complicated clinical management. The World Health Organization (WHO) has classified pneumococcal diseases as conditions requiring “extremely high priority” for prevention through vaccination. Immunization against pneumococcal disease represents the most cost-effective and efficient method for preventing infection caused by Streptococcus pneumonia [12]. Currently, pneumococcal vaccines approved for use in China are primarily divided into two categories: polysaccharide vaccines and pneumococcal conjugate vaccines. The 23-valent pneumococcal polysaccharide vaccine (PPSV23) is predominantly recommended for individuals aged 60 years and older, high-risk individuals aged two years and above with chronic underlying conditions such as cardiovascular disease, diabetes, and chronic respiratory diseases, as well as those with compromised immune systems. Pfizer’s 7-valent pneumococcal conjugate vaccine (PCV7) and 13-valent pneumococcal conjugate vaccine (PCV13) were officially approved for the Chinese market in 2008 and 2016. PCVs are primarily intended to prevent pneumococcal diseases in infants and children aged 2 months to 5 years, respectively. From 2019 to 2024, PCV13 vaccines developed by companies such as Walvax Biotechnology Co., Ltd.,Kunming, China, Beijing Minhai Biotechnology Co., Ltd.,Beijing, China, and CanSino Biologics Inc.,Tianjin, Chinawere successively approved for market authorization. Globally, 168 countries and regions have incorporated PCVs into their national immunization programs (NIPs) [13]. However, in China, pneumococcal vaccines remain categorized as non-immunization program vaccines and have not yet been included in the NIPs. However, widespread PCV use has also raised concerns about serotype replacement, wherein non-vaccine serotypes may fill the ecological niche left by vaccine-targeted strains [14,15]. This underscores the need for robust and ongoing epidemiological surveillance to guide vaccine updates and optimize immunization strategies.

A critical component of evaluating any vaccine’s effectiveness is understanding its performance relative to baseline—or naturally acquired—antibody levels in the target population [16]. Determining these baseline levels provides insights into natural immunity, informs thresholds for protective antibody concentrations, and supports decisions regarding the need for vaccination or revaccination [17,18]. Specifically, characterizing the distribution of serotype-specific IgG levels in unvaccinated populations allows for better interpretation of vaccine-induced responses and disease risk associated with particular serotypes.

Despite the availability of surveillance data on pneumococcal serotype distribution and antibiotic resistance from local Centers for Disease Control and Prevention (CDCs) and hospitals in China, comprehensive data on serotype-specific baseline IgG levels in unvaccinated healthy individuals remain lacking [19,20,21]. Given China’s substantial pneumococcal disease burden [22,23], filling this knowledge gap is essential for designing effective vaccination policies. While several countries have conducted baseline studies on pneumococcal serotype-specific antibodies [17,24,25], such data are currently unavailable at the national level in China.

To address this need, we conducted a large-scale, multicenter retrospective study to assess baseline IgG antibody levels against 13 pneumococcal capsular serotypes (1, 3, 4, 5, 6A, 6B, 7F, 9V, 14, 18C, 19A, 19F, and 23F) in a healthy Chinese pediatric population. This research represents the first nationwide baseline serological survey of pneumococcal IgG antibodies in China, providing a foundational database for vaccine developers and policymakers. The findings aim to inform regional immunization strategies, guide the prioritization of PCV13 deployment, and support regulatory decisions regarding the potential inclusion of PCV13 in China’s NIPs.

## 2. Materials and Methods

### 2.1. Serum Samples

A total of 21,265 pre-immunization serum samples were collected from healthy children aged 2 months (minimum 6 weeks) to 5 years (before their sixth birthday) across eight provinces in four major geographical regions of China from 2016 to 2023. These samples were derived exclusively from clinical trials designed to assess the immunogenicity and safety of PCV13; consequently, the serum samples reflect the cohorts enrolled in these specific trials. These trials spanned from 2016 to 2023 and encompassed diverse regions across mainland China, potentially enhancing the representativeness of the study cohort. The regional distribution was as follows: 11,251 from North China, including 8193 (4104 males, 4089 females) from Henan, 2138 (1057 males, 1081 females) from Shanxi, 837 (450 males, 387 females) from Hebei, and 83 (40 males, 43 females) from Beijing; 2641 from East China, including 2244 (1178 males, 1066 females) from Jiangsu and 397 (200 males, 197 females) from Zhejiang; 3595 (1841 males, 1754 females) from South China (Guangxi); and 3778 (2014 males, 1764 females) from Southwest China (Yunnan). Of the total participants, 10,884 were male and 10,381 were female.

According to national PCV immunization schedules, children were grouped by age into five cohorts: 2 months, 3–6 months, 7–11 months, 12–23 months, and 2–5 years. None of the participants had received PCV or PPSV prior to enrollment, nor had they experienced febrile or infectious illness within seven days before sample collection. Blood samples were collected via venipuncture on the day of enrollment, and serum was separated and stored at below –20 °C until analysis. The international reference serum 007sp was obtained from the National Institute for Biological Standards and Control (UK), and quality control serum 09CS was provided by Lanzhou Institute of Biological Products Co., Ltd., Lanzhou, China. Sample collection spanned seven years (2016–2023), with serum specimens obtained from regions including North China, East China, South China, and Southwest China. During this period (2016–2023), PCVs were not included in China’s Expanded Program on Immunization (EPI), and no pneumococcal disease outbreaks were recorded nationally.

### 2.2. Pneumococcal Polysaccharide Antigen

Polysaccharide antigens corresponding to the 13 PCV serotypes (1, 3, 4, 5, 6A, 6B, 7F, 9V, 14, 18C, 19A, 19F, and 23F) were provided by Yunnan Walvax Biotechnology Co., Ltd.,Kunming, China, Lanzhou Institute of Biological Products Co., Ltd.,Lanzhou, China, Beijing Minhai Biotechnology Co., Ltd., Beijing, China, and Beijing Zhifei Lvzhu Biopharmaceutical Co., Ltd., Beijing, China. All antigens complied with the quality standards specified in the 2020 edition of the Chinese Pharmacopeia and the ninth edition of the European Pharmacopeia. Antigens were pooled in equal amounts and stored by the China Institute for Food and Drug Control for further use.

### 2.3. Serotype-Specific IgG Antibody ELISA

The assay was based on the WHO-recommended quantitative ELISA protocol (2004) for measuring serotype-specific IgG antibodies against pneumococcal capsular polysaccharides. This method was validated by the National Institute for Food and Drug Control and is widely adopted globally.

Each serotype-specific antigen was coated onto 96-well ELISA plates (100 μL/well) and incubated at room temperature overnight, sealed, and stored in a moist box at 2–8 °C. CWPS and 22F polysaccharides were used to absorb non-specific antibodies, prepared as 5 μg/mL in dilution buffer. Samples, 007sp, and 09CS were pre-absorbed and serially diluted (2.5-fold across seven rows; row H served as blank). ELISA operations were performed on the Tecan Freedom EVO 150 platform (Tecan, Mannedorf, Switzerland) following WHO procedures.

Plates were washed three times with TBS buffer, and 50 μL/well of diluted samples were added and incubated for 2 h at room temperature. After washing, 100 μL/well of alkaline phosphatase-conjugated goat anti-human IgG (1:40,000 dilution, sigma, St. Louis, MI, USA) was added and incubated for another 2 h. Following another wash, 100 μL/well of p-nitrophenyl phosphate (PNPP, 1 mg/mL, sigma, St. Louis, MI, USA) was added and incubated for color development. The reaction was stopped with 50 μL/well of 3 mol/L NaOH, and absorbance was measured at 405–690 nm using a Tecan Sunrise ELISA reader(Tecan, Salzburg, Austria).

### 2.4. Statistical Analysis

Samples were analyzed by region (North, East, South, Southwest), age group (2 months, 3–6 months, 7–11 months, 12–23 months, and 2–5 years), and gender. The primary endpoints included the geometric mean concentration (GMC) of serotype-specific IgG and the proportion of individuals with antibody levels ≥0.35 μg/mL (defined as seropositive).

Statistical analysis was conducted in several steps:(1)Descriptive Statistics: Overall distribution of IgG levels was summarized nationwide without stratification.(2)Two-Way ANOVA: Baseline antibody levels (log-transformed) were assessed for fixed effects of region, age, and gender.(3)Pairwise Comparisons: GMC and seropositivity by region; region-stratified comparisons by age group; GMC and seropositivity by age group; age-stratified comparisons across regions.(4)Gender-Based Comparisons: National-level GMC and seropositivity differences between males and females; region-stratified comparisons by gender; age group-stratified comparisons by gender.

Measurement data were described using mean, median, standard deviation, and range, while categorical data were reported as frequency and percentage. Statistical analyses were performed using SAS 9.4 and GraphPad 9.5. For seropositivity, 95% confidence intervals were estimated using the Clopper–Pearson method, and group differences were assessed using the χ^2^ test. For GMC, confidence intervals were calculated, and differences among multiple groups were analyzed using One-Way ANOVA. Log-transformed Student’s *t*-tests were used for pairwise group comparisons. A two-sided *p* < 0.05 was considered statistically significant.

### 2.5. Ethical Approval

This study was conducted in collaboration with six vaccine development enterprises and approved by the relevant ethics committees prior to initiation of any clinical activity. Written informed consent was obtained from all participants. Approvals were granted by the following: Medical Ethics Committee of Henan Provincial Center for Disease Control and Prevention (Case Number: 2015-YM-004-01, Acceptance Date: 12 October 2015; Case Number: 2018-YM-006-02, Acceptance Date: 26 December 2018; Case Number: 2019-YM-006-02, Acceptance Date: 17 July 2019); Ethics Review Committee of Guangxi Zhuang Autonomous Region Center for Disease Control and Prevention (Case Number: GXIRB2021-0035-1, Acceptance Date: 19 September 2022); Ethics Committee for Vaccine Clinical Trials of Yunnan Provincial Center for Disease Control and Prevention (Case Number: 2021-12, Acceptance Date: 21 September 2021); Ethics Review Committee of Jiangsu Provincial Center for Disease Control and Prevention (Case Number: JSJK2015-A002-02, Acceptance Date: 27 March 2015); Clinical Trial Ethics Review Committee of Zhejiang Provincial Center for Disease Control and Prevention (Case Number: 2020-001-01, Acceptance Date: 20 October 2020); Ethics Committee of Chaoyang District Center for Disease Control and Prevention (Case Number: CYCDPCIRB-20150804-1, Acceptance Date: 4 September 2022); Ethics Committee of Hebei Provincial Center for Disease Control and Prevention (Case Number: IRB2015-003, Acceptance Date: 9 September 2015); Ethics Review Committee of Shanxi Provincial Center for Disease Control and Prevention (Case Number: SXCDCIRBPJ201500401, Acceptance Date: 12 August 2015).

## 3. Results

### 3.1. Demographic Characteristics and Geographical Distribution

A total of 21,265 serum samples were collected, including 10,884 males and 10,381 females, with a male-to-female ratio of 1.05:1. The samples were distributed across four major geographical regions in China: North, East, South, and Southwest. According to the Hu Huanyong Line (Figure 1), a demographic boundary that separates densely and sparsely populated regions in China, the majority of samples were from southeastern regions with high population density. Based on the 2020 Seventh National Census [26], Southeastern regions accounted for approximately 94% of the national population, ensuring high population representativeness.

Participants were categorized into five age groups: 2 months, 3–6 months, 7–11 months, 12–23 months, and 2–5 years (Table 1 and Table 2). Sample sizes were relatively balanced across age groups: 32.7% (2 months), 26.7% (3–6 months), 8.3% (7–11 months), 12.1% (12–23 months), and 20.1% (2–5 years). Geographically, the majority were from North China (53.0%), followed by Southwest (17.7%), South (16.9%), and East China (12.4%). Figure 1 shows the geographic distribution of participants.

### 3.2. Nationwide Overview of Baseline Antibody Levels

In total, 276,445 valid antibody measurements were obtained. As shown in Figure 2, baseline GMCs for serotypes 6B, 14, 19A, and 19F exceeded the protective threshold of 0.35 μg/mL, with a strong correlation observed between GMCs and seropositive rates. GMC profiles were similar between males and females, and the seropositivity curves largely overlapped.

The positive rates between genders were compared using the χ^2^ test, which revealed significant differences for most serotypes except serotype 4. Geometric mean concentrations (GMCs) between genders were analyzed using *t*-tests on log-transformed data and showed significant differences across all serotypes. However, when the data were further stratified by geographic region or age group, gender differences in most serotypes were no longer statistically significant.

Across all regions, serotype-specific GMCs were highly similar between males and females, and the distribution curves of seropositivity rates showed clear overlap. In subgroup analyses stratified by region, the *p*-values for various serotypes indicated that gender had little effect on overall GMC levels. Similarly, when stratified by age group, the GMCs and seropositivity rate curves of males and females remained largely overlapping, further confirming the minimal impact of gender on antibody levels.

In all regions and age groups, serotypes 6B, 14, and 19F consistently showed higher GMCs and positive rates in both males and females, while serotypes 3, 4, and 5 remained lower. These patterns were consistent with the overall study population. The findings suggest that gender does not have a meaningful influence on serotype-specific antibody levels, regardless of region or age group.

When stratified by region, subjects from South China exhibited higher GMC levels across most serotypes. Age-stratified analysis revealed a U-shaped distribution of baseline antibody levels: high in infants at 2 months (due to maternal antibodies), decreasing in mid-infancy (due to waning maternal antibodies and immature immune responses), and rising again in children aged 2–5 years (due to natural exposure and immune maturation) [27].

#### 3.2.1. Regional Baseline GMC and Seropositivity Without Age Stratification

As shown in Figure 3, the regional GMC trends across the 13 serotypes were broadly consistent. Serotypes 14, 19A, 19F, and 6B had higher GMCs across all regions. However, absolute GMC values varied significantly between regions (*p* < 0.0001), as did seropositivity rates for several serotypes, such as types 4 and 5.

#### 3.2.2. Regional Baseline GMC and Seropositivity with Age Stratification

Figure 4 illustrates the baseline antibody levels and seropositivity rates across age groups and regions. In general, in the 2-month group, GMCs for 14, 18C, 19F, and 6B were ≥0.35 μg/mL across all regions, while serotypes 1, 3, 4, and 5 had the lowest GMCs. The Southwest region showed the highest antibody levels. In the 3–6-month group, 14, 18C, and 19F again showed the highest levels. The East region had the highest GMCs. For 7–11 months, only North and East China were represented. Serotypes 14, 18C, and 19F remained above the threshold, with North China showing higher GMCs. In 12–23 months, three regions were represented; serotypes 14, 18C, and 19F again exceeded 0.35 μg/mL, with East China having the highest overall levels. In 2–5-year-olds, all serotypes in all regions showed elevated GMCs, especially for 14, 19A, 19F, 23F, and 6B. North China had the highest values in this age group.

Overall, GMCs for 14, 18C, and 19F consistently exceeded 0.35 μg/mL across all ages and regions. Serotypes 6B, 23F, and 19A also showed high levels in specific age groups. Conversely, 1, 3, 4, 5, and 9V had consistently low GMCs. Regional heterogeneity was evident, confirming age- and geography-related variation in pneumococcal antibody levels.

### 3.3. Baseline GMC and Positive Rate Analysis Across Age Groups

#### 3.3.1. Overview of Baseline Antibody Levels and Positive Rates Across Five Age Groups

Figure 5 illustrates the distribution of GMC and positive rates across the five age groups. A characteristic U-shaped distribution in GMC levels was observed for all 13 pneumococcal serotypes: antibody levels decreased from 2 months to 7–11 months, then increased gradually through 12–23 months, peaking at 2–5 years. This trend was consistent across most serotypes and showed statistically significant differences among age groups (*p* < 0.0001). The positive rate trends mirrored those of GMC, further confirming the strong correlation between GMC values and positivity thresholds for each serotype.

#### 3.3.2. Baseline Antibody Levels and Positive Rates Across Age Groups Stratified by Region

Figure 6 presents a stratified analysis of GMC and positive rates in the five age groups across North, East, South, and Southwest China. In North China, most serotypes demonstrated a U-shaped GMC distribution, with exceptions (e.g., types 3, 4, 5A, 7F, and 23F) showing no significant age-related differences. Antibodies for types 14, 19A, and 19F consistently exceeded the protection threshold (≥0.35 μg/mL) in all age groups. Types 4 and 3 remained at relatively low levels throughout. The GMC of 9V and 18C was particularly low at 7–11 months. In East China, U-shaped distributions were also seen, with some serotypes (7F, 9V, 14, 18C, and 19F) showing no significant age-based differences. GMC values of types 14, 19A, and 19F remained high across all age groups. Serotypes 4 and 3 were persistently low, and types 9V, 18C, 4, and 7F were significantly lower at 7–11 months. In South China, only age groups at 2 months, 3–6 months, and 2–5 years were included. GMC levels showed a U-shaped distribution across these three groups. Most serotypes (excluding 18C) had significantly different levels between groups. Type 14, 19A, and 19F GMCs were consistently above the protection threshold. Notably, 9V, 5, 4, and 3 types were lowest at 3–6 months. In Southwest China, data were available for age groups 2 months, 3–6 months, 12–23 months, and 2–5 years. The U-shaped pattern persisted, though for some serotypes (e.g., types 5, 6A, 6B, 19A, 19F, and 23F), age-based differences were not statistically significant. Types 9V and 18C showed the lowest GMCs at 3–6 months.

In summary, U-shaped distribution across all regions and most serotypes indicates a nadir in antibody levels during infancy (particularly at 3–11 months), identifying a vulnerable immunological window. Types 14, 19A, and 19F consistently showed GMCs above the protection threshold in all ages and regions, while types 3 and 4 were persistently lower. Region-specific trends were observed in the baseline antibody levels across different age groups. In North China, the serotypes 9V and 18C exhibited particularly low GMC levels in infants aged 7–11 months. In South China, lower antibody levels were noted for serotypes 9V, 5, 4, and 3, especially in the 3–6-month age group. Similarly, in Southwest China, serotypes 9V and 18C showed reduced GMC levels among infants aged 3–6 months.

## 4. Discussion

Baseline antibody levels are crucial for accurately assessing vaccine immunogenicity. While numerous domestic and international studies have explored the immunogenicity of pneumococcal vaccines, few have addressed the regional distribution of baseline IgG levels. To our knowledge, this is the first large-scale, cross-sectional, multicenter study in China investigating naturally acquired IgG antibody levels against the 13 serotypes included in PCV13 in healthy, vaccine-naïve individuals of specific age groups. These baseline levels reflect natural immunity acquired through asymptomatic carriage or past infections, and are instrumental in optimizing serotype composition in vaccine formulations. Understanding baseline antibody distributions not only enhances knowledge of natural immunity but also informs vaccine design, efficacy evaluation, and immunization planning [24]. These results provide important evidence to inform the incorporation of PCV13 into China’s national immunization strategy and to support the formulation of optimized, region-specific vaccination policies.

The human immune system can generate natural or adaptive antibodies in response to antigen exposure, even without vaccination, though these differ in specificity and function. As Tanaka et al. [28] suggested, establishing pre-vaccination IgG reference ranges helps elucidate the incidence of serotype-specific IPD, baseline immune responsiveness, and the necessity of re-vaccination. This study provides large-scale data from major population centers in China, representing the general natural immune landscape and laying the foundation for a national pneumococcal immunity database.

This study revealed several notable findings: First, serotypes 6B, 14, 19A, and 19F showed the highest baseline GMCs (all > 0.45 μg/mL) nationwide, aligning with prior reports [29]. Notably, serotype 14 exhibited the highest GMC (1.64 μg/mL), while serotypes 3 and 4 had GMCs far below the 0.35 μg/mL protective threshold, suggesting weak immunogenicity and insufficient herd immunity. Global data indicate that serotype prevalence varies by region; for example, serotypes 14, 6B, 19F, 18C, 4, and 23F are common in North America [30,31]; in Western Europe, serotypes 14, 19F, 6B, 18C, 1, 9V, and 23F predominate [32]; Thailand reports 6B, 23F, and 14 [33]; Malaysia 14, 6B, 19A, and 6A [34]; Mexico 19A, 3, and 19F [35]; and northern Russia 19F, 23F, and 6A [36]. Serotype distributions are dynamic over time. For instance, serotypes 2 and 5, dominant in the 1940s in North America and Europe, are now rare there but still prevalent in South America [37]. In our study, 9 of 13 serotypes had GMCs exceeding 0.35 μg/mL (1, 6A, 6B, 7F, 9V, 14, 19A, 19F, and 23F). However, this threshold originates from early clinical trials, and its direct correlation with clinical protection remains debated. Second, gender analysis showed highly similar GMC patterns and overlapping positive rate curves between males and females. Although multivariate ANOVA indicated significant gender effects on some serotypes, these differences diminished substantially in subgroup analyses by age and region. The marginally higher GMCs in females were not statistically meaningful, contrasting with previous reports where males showed higher baseline levels [38,39]. Ulanova et al. [17] similarly found no significant gender effect. Hence, while large sample sizes may produce statistical significance, the clinical relevance of gender differences in baseline IgG is limited and must be interpreted within broader demographic and health contexts. Third, while overall spatial heterogeneity was limited, regional differences were evident. South China had the highest GMC and positive rates, followed by East China, with North China showing the lowest. This may relate to regional epidemiology; notably, South China had exceptionally high GMCs for serotype 14 (3.7 μg/mL), suggesting a stronger natural exposure and possible need for region-specific immunization strategies. Fourth, a U-shaped distribution of baseline GMCs across age groups was observed, with lower levels in early infancy and increases with age. Excluding maternal antibody interference, IgG concentrations showed a positive correlation with age, and older children exhibited broader serotype coverage above the protective threshold. Serotypes 14, 19A, and 19F consistently exceeded 0.35 μg/mL in all age groups, while types 3 and 4 remained low across the board. Notably, serotype-specific antibody troughs were found at 3 months and 7–11 months—critical immune vulnerability windows. Region-specific age effects were also detected. For example, in North China, antibodies against 9V and 18C dropped markedly at 7–11 months. In South China, antibodies against 9V, 5, 4, and 3 were lowest at 3–6 months. Similarly, in Southwest China, antibodies against 9V and 18C were lowest in infants aged 3–6 months. These patterns likely reflect varying regional exposure and serotype circulation.

This study has several limitations. First, the cross-sectional design is unable to capture dynamic changes in antibody levels after vaccination. Future longitudinal studies are necessary to assess how baseline differences affect vaccine responses, with further analysis of baseline antibody levels across more vaccine serotypes to be conducted. Second, the scope of this study did not address emerging non-vaccine serotypes, which require attention to support long-term vaccine strategies.

## 5. Conclusions

This nationwide multicenter study provides the first comprehensive baseline data on serotype-specific IgG levels against PCV13 serotypes in unvaccinated healthy Chinese children. The findings reveal substantial regional and age-related variation, with consistently high levels for serotypes 14, 19A, and 19F, and low levels for serotypes 3 and 4. A notable antibody nadir between 3 and 11 months suggests a critical period of vulnerability, reinforcing the need for timely vaccination.

## Figures and Tables

**Figure 1 vaccines-13-00847-f001:**
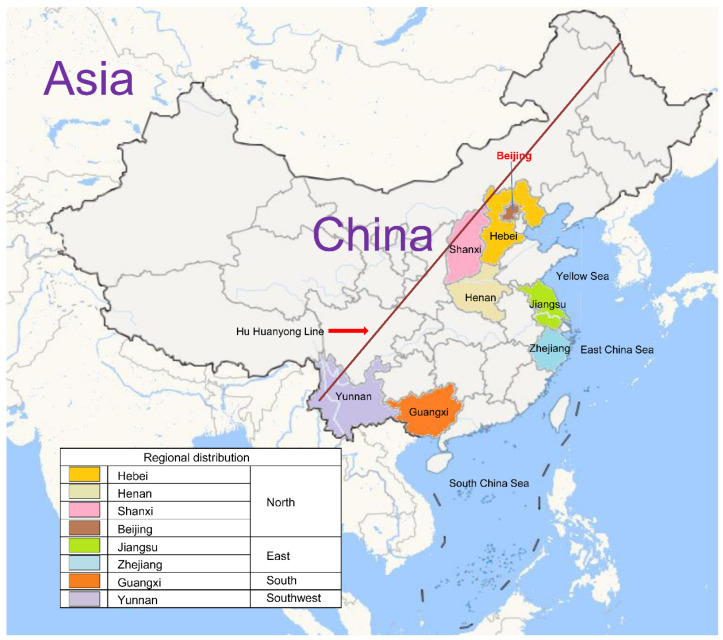
Geographic distribution of study participants. According to the Hu Huanyong Line, a demographic boundary that separates densely and sparsely populated regions in China, the majority of samples were collected from the southeastern region, where the population density is highest.

**Figure 2 vaccines-13-00847-f002:**
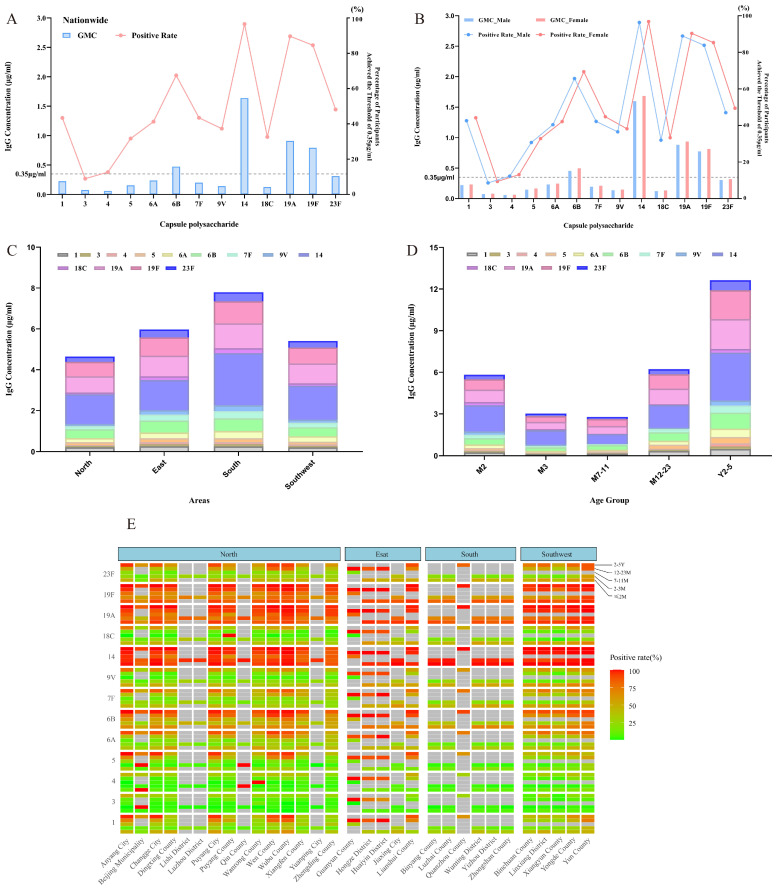
National overview of baseline pneumococcal IgG antibody levels. (**A**) National baseline geometric mean concentrations (GMCs) and seropositivity rates. (**B**) National baseline GMCs and seropositivity rates by gender. (**C**) GMC stacking charts for four geographic regions. (**D**) GMC stacking charts for five age groups. (**E**) Heat map of seropositivity rates across counties and cities, stratified by region and age group. For each serotype, five age groups are represented per region. The color gradient from green to red indicates increasing seropositivity rates. Gray indicates that no samples were available for that age group in a given region.

**Figure 3 vaccines-13-00847-f003:**
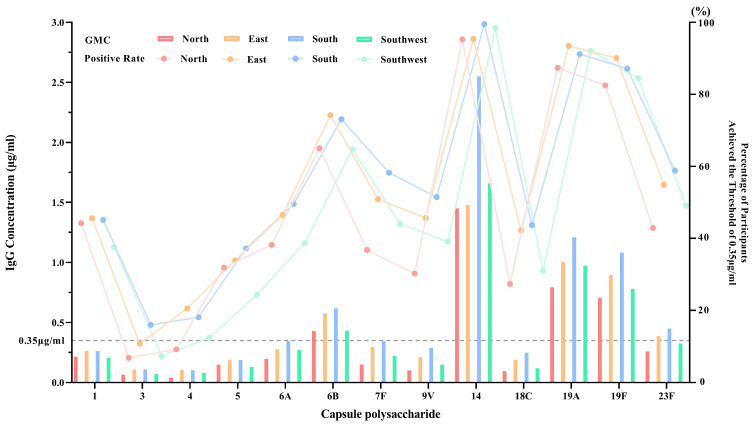
Regional distribution of baseline geometric mean concentrations (GMCs) and seropositivity rates. The bar chart represents GMCs, and the line chart shows seropositivity rates for each region. The *X*-axis indicates pneumococcal serotypes; the left *Y*-axis represents GMC values, and the right *Y*-axis represents the proportion of samples with GMCs above 0.35 μg/mL (seropositivity rate).

**Figure 4 vaccines-13-00847-f004:**
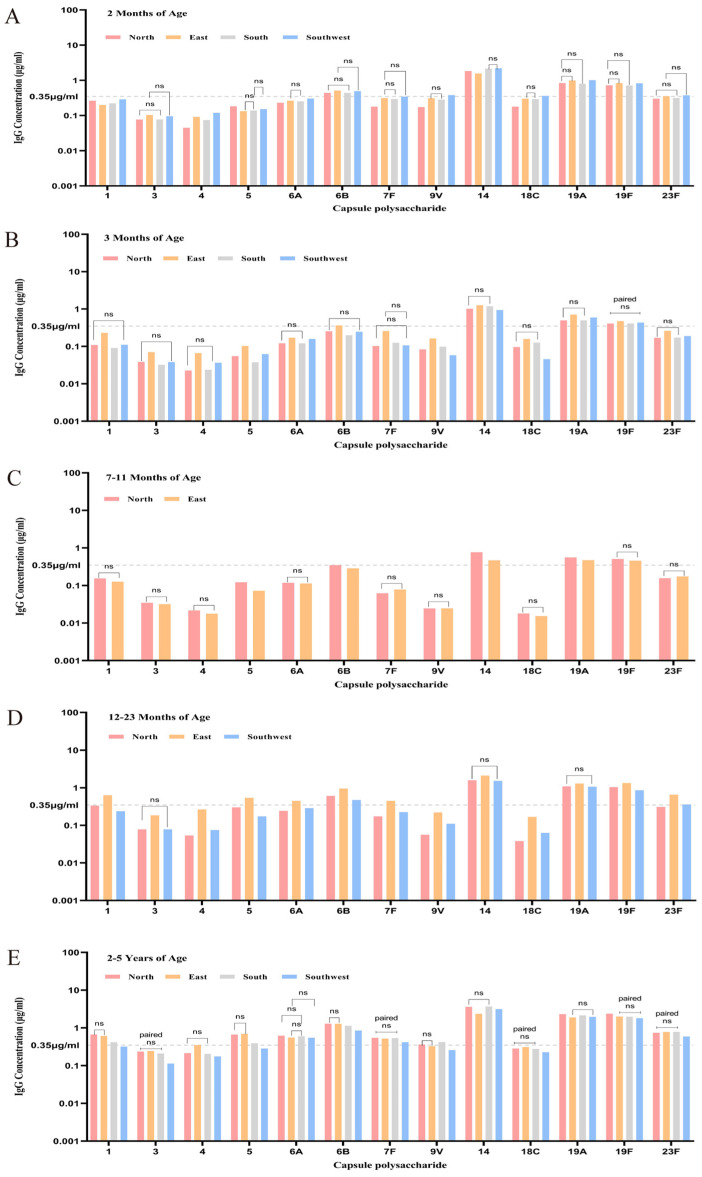
Age- and region-specific distribution of baseline GMCs and seropositivity rates. (“-----” represents the seropositivity threshold of 0.35 μg/mL. “ns” indicates no statistically significant difference among the four regions within the corresponding age group). (**A**) Comparative analysis of 13 serotypes in 2-month-old children across four regions. (**B**) Comparative analysis of 13 serotypes in 3-month-old children across four regions. (**C**) Comparative analysis of 13 serotypes in children aged 7–11 months across four regions. (**D**) Comparative analysis of 13 serotypes in children aged 12–23 months across four regions. (**E**) Comparative analysis of 13 serotypes in children aged 2–5 years across four regions.

**Figure 5 vaccines-13-00847-f005:**
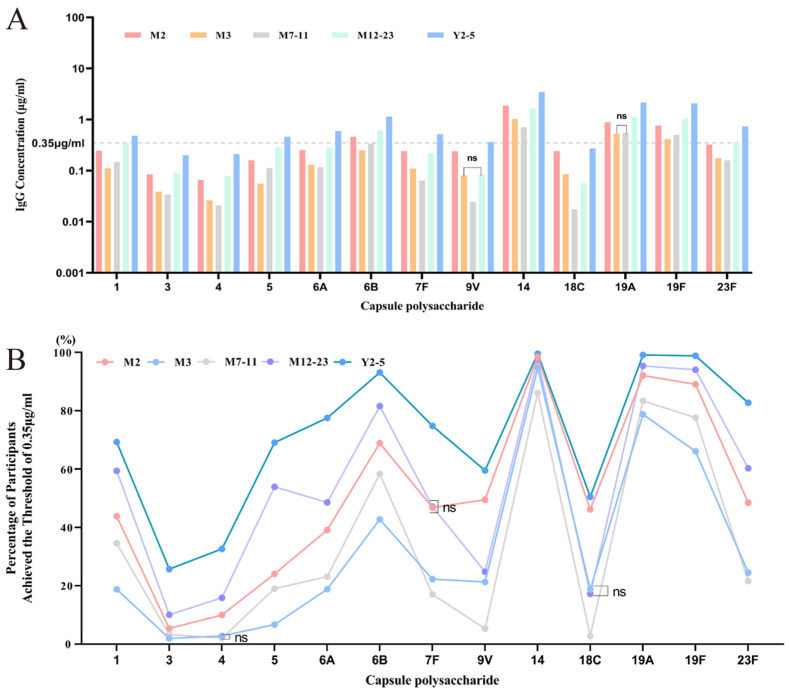
Overview of baseline GMC and positive rate of subjects in five age groups. (**A**) Baseline GMCs in five age groups. The horizontal line represents the seropositivity threshold of 0.35 µg/mL. “ns” indicates that there is no statistically significant difference among the four regions within the corresponding age group. (**B**) Seropositivity rates in five age groups. The *X*-axis represents pneumococcal serotypes, and the *Y*-axis represents the proportion of subjects with GMCs above 0.35 μg/mL (positive rate).

**Figure 6 vaccines-13-00847-f006:**
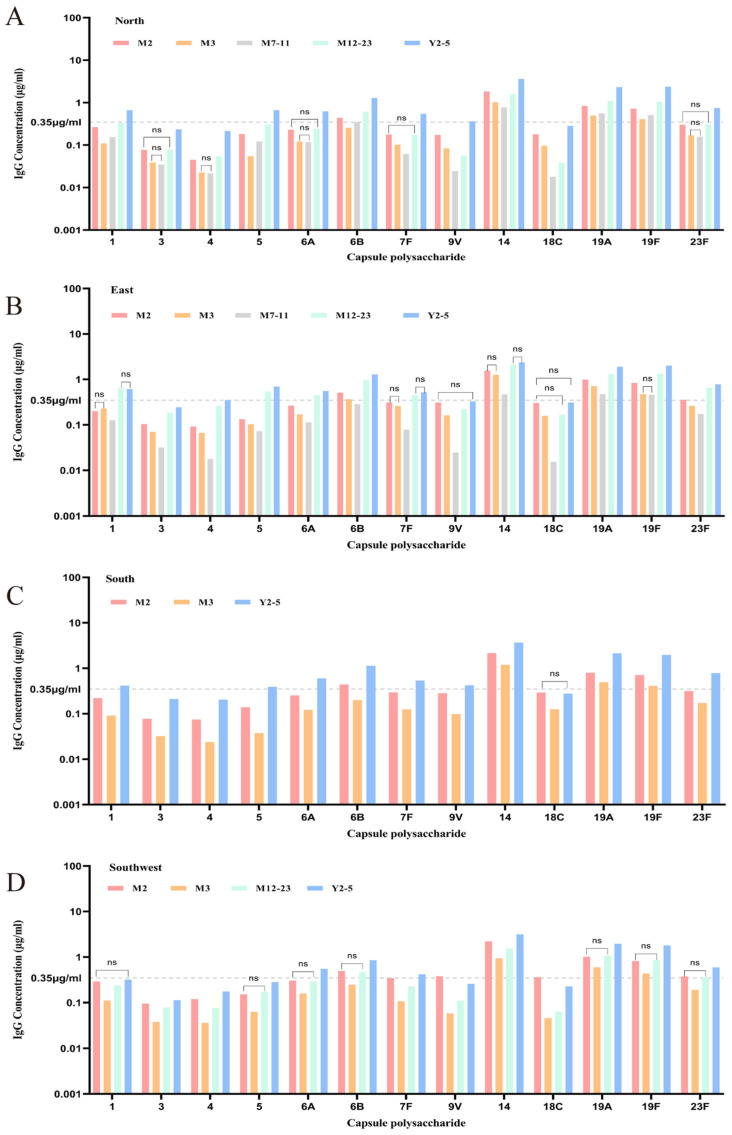
Overview of baseline GMC and positive rates in five age groups in four regions. (“-----” indicates the seropositivity threshold of 0.35 μg/mL. “ns” denotes that there is no statistically significant difference among the four regions within the corresponding age group). (**A**) Baseline GMCs by age group in North China. (**B**) Baseline GMCs by age group in East China. (**C**) Baseline GMCs by age group in South China. (**D**) Baseline GMCs by age group in Southwest China.

**Table 1 vaccines-13-00847-t001:** Demographic characteristics of healthy participants not vaccinated with PCV13.

Age Groups	Number of Subjects	Male	Female	Male-to-Female Ratio
2 months	6965	3570	3395	1.05:1
3–6 months	5675	2598	2477	1.05:1
7–11 months	1769	875	894	0.98:1
12–23 months	2581	1344	1237	1.09:1
2–5 years	4275	2183	2092	1.04:1
Total	21,265	10,884	10,381	1.05:1

**Table 2 vaccines-13-00847-t002:** Regional distribution of serum samples collected before primary immunization across China.

Geographical Region	Province/Municipality	Sample Time	Number of Subjects		Male	Female	Total Population (2020)	Sampling Ratio (‱)
North China	Henan Province	2016 2019 2020 2021	1072 1793 2007 3321	8193	4104	4089	99,365,519	0.82
	Shanxi Province	2016 2022	839 1299	2138	1057	1081	34,915,616	0.61
	Hebei Province	2016	837	837	450	387	74,610,235	0.11
	Beijing Municipality	2016	83	83	40	43	21,893,095	0.04
East China	Jiangsu Province	2017 2018	1200 1044	2244	1178	1066	84,748,016	0.26
	Zhejiang Province	2020 2021	43 354	397	200	197	64,567,588	0.06
South China	Guangxi Zhuang Autonomous Region	2022	3595	3595	1841	1754	50,126,804	0.72
Southwest China	Yunnan Province	2023	3778	3778	2014	1764	47,209,277	0.80

Note: ‱ indicates that the sampling ratio is one in ten thousand.

## Data Availability

The data that support the findings of this study are available from the corresponding authors upon reasonable request.

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
