# Peer review of "Baseline Analysis of Serotype-Specific IgG Antibody Levels for 13-Valent Pneumococcal Conjugate Vaccine in Healthy Chinese Individuals: A Multicenter Retrospective Study"

_vaccines, 2025, doi:10.3390/vaccines13080847_

Round 1
Reviewer 1 Report
Comments and Suggestions for Authors
Baseline analysis of serotype specific antibodies for 13-valent pneumococcal serotypes in China
A study testing many participants from multiple provinces from China – would be a great study if below and many additional aspects were improved to present and discuss the results meaningfully. I hope the other reviewers will also assist.
Very important in abstract and definitely in the introduction to talk about any pneumococcal vaccines in use in China – both pure polysaccharide and conjugate – whether used in elderly and at risk adolescents and elderly (polysaccharide) and when routine PCV was introduced and how (schedule and any catch up etc).
All figures and tables need proper titles, to help them stand alone: time/place/person and then describing Y and X axes properly and thinking for each figure/table what data is being displayed.
Line 49: rewrite sentence pneumococcus is the only cause of IPD, it cannot be the “predominant cause”. IPD is a specific diagnosis, CAP is a syndrome.
Line 63 64: and this is throughout the paper when referencing other papers – is to give n/N to help in the interpretation of the % coted.
Line 83: write out CDC
Line 93, and then in methods 102 to 119: please tell us how the stratified sampling was done: random sampling at the population level, or convenience sampling – just give us all the details to help understand what the results may mean.
Also important is to place the specimens meaningfully in time – so over the 7-year period – how many specimens from which regions were taken in which years. A 7-year period is a long time – and although possible to analyse in a cross-sectional way – you need to convince us of no known pneumococcal outbreaks in the regions from which the individuals came and that vaccination strategies did not change.
Line 197/198: immediate refer to figure 1 to help the reader.
Line 221 – what is the measure of the statistical significance – a quantitative measure is needed, confidence intervals and then a result for the statistical test etc etc.
Table 2: it would be good to know the total population for each province, then the number of subjects – just to see if relative proportions of each province. I am not sure if the years can be added here. See where it fits best.
Figure 2 C and D are not meaningful?? By stacking the columns there is no data by serotype (the data are not addidative!!) – please redo these figures to show the data meaningfully. So Line 224 to 228 will need to be redone once looking at data more carefully (it contradicts figure 3!). Also define the positive rate is the cut off of >0.35 – so the figures can stand alone.
Figure 2E: what does the grey colour mean – it is not in the key?
Figure 4 is too small. And needs a careful title, and check how it differs from figure 5. It would be better to choose how best to show difference by age, but not do it in multiple ways. Also see Figure 6. These figures are repetitive and do not help the reader. Rather do two good figures, visible and large enough and carefully interpret the result.
Figure 7 is again by gender – so links to figure 2 and then work out if differences by gender, and whether truly significant – if not, no need to analyse in such detail. Also see figure 8.
The reader is being flooded by figures – when many fewer would tell the same story – some of the additional analyses could go into supplementary materials.
Discussion: line 331 – 340 – baseline antibody data are useful but not “crucial” for assessing vaccine immunogenicity, and also do not reflect disease – so may not be so helpful in vaccine design and efficacy evaluations – this paragraph would benefit from a more careful discussion – also the whole 1st paragraph says nothing about what the study showed, but are rather sweeping statements with no references.
The discussion would benefit from paragraphs in themes – comparing the study results to international findings with references.
A limitation would be the 7-year period – as much happens in 7 years.
Line 391 – why bring in immunogenicity (vaccine response?? Throughout the paper differentiate when discussing antibody response to natural infection vs vaccination, also define carefully when referencing the literature) against serotype 3 – as this was not at all studied in this paper – rather discuss what we know about responses to infection with serotype 3. Your discussion and comparison to the literature need to look at natural infection responses.
Author Response
Comments and Suggestions for Authors
Baseline Analysis of Serotype-Specific IgG Antibody Levels for 13-Valent Pneumococcal Conjugate Vaccine in Healthy Chinese Individuals: A Multicenter Retrospective Study
A study testing many participants from multiple provinces from China – would be a great study if below and many additional aspects were improved to present and discuss the results meaningfully. I hope the other reviewers will also assist.
At the outset, we wish to convey our heartfelt appreciation to the reviewers for the time and care they have devoted to scrutinizing our manuscript, as well as insightful and constructive feedback, which have significantly enhanced the rigor and clarity of manuscript.
- Very important in abstract and definitely in the introduction to talk about any pneumococcal vaccines in use in China – both pure polysaccharide and conjugate – whether used in elderly and at risk adolescents and elderly (polysaccharide) and when routine PCV was introduced and how (schedule and any catch up etc).
Response: Thank you for your insightful suggestion.We have clarified the description of pneumococcal vaccine application background in China within the Introduction section, as :WHO has classified pneumococcal diseases as conditions requiring "extremely high priority" for prevention through vaccination, Immunization against pneumococcal disease represents the most cost-effective and efficient method for preventing infection caused by Streptococcus pneumonia [12]. Currently, pneumococcal vaccines approved for use in China are primarily divided into two categories: polysaccharide vaccines and pneumococcal conjugate vaccines. 23-valent pneumococcal polysaccharide vaccine (PPSV23) is predominantly recommended for individuals aged 60 years and older, high-risk individuals aged two years and above with chronic underlying conditions such as cardiovascular disease, diabetes, and chronic respiratory diseases, as well as those with compromised immune systems. Pfizer's 7-valent pneumococcal conjugate vaccine(PCV7) and 13-valent pneumococcal conjugate vaccine (PCV13) were officially approved for the Chinese market in 2008 and 2016. PCV are primarily intended for prevent pneumococcal diseases in infants and children aged 2 months to 5 years.respectively. From 2019 to 2024, PCV13 vaccines developed by companies such as Walvax Biotechnology Co., Ltd, Beijing Minhai Biotechnology Co., Ltd and CanSino Biologics Inc. were successively approved for market authorization. Globally, 168 countries and regions have incorporated PCVs into their national immunization programs (NIPs) [13]. However, in China, pneumococcal vaccines remain categorized as non-immunization program vaccines and have not yet been included in the NIPs.(Page 2, Lines 69-88)
- All figures and tables need proper titles, to help them stand alone: time/place/person and then describing Y and X axes properly and thinking for each figure/table what data is being displayed.
Response: We fully acknowledge the importance of making all figures and tables self-explanatory by including clear titles, properly labeled axes, and sufficient contextual information. In response to your comment, we have revised the titles and annotations accordingly, including adding titles within subgraphs where applicable. We have ensured that each figure and table can be understood independently of the main text, providing necessary context without redundancy. These revisions improve the clarity, readability, and overall compliance with scientific publishing standards.
For example, the following picture is the title and explanation of the small picture in Figure 2 of the manuscript.
- Line 49: rewrite sentence pneumococcus is the only cause of IPD, it cannot be the “predominant cause”. IPD is a specific diagnosis, CAP is a syndrome.
Response: We fully agree with your correction and acknowledge the inaccuracy in our original statement. In the revised manuscript, we have updated the sentence on Page 2, Lines 48-50. The revised sentence emphasizes that IPD is a specific diagnosis caused exclusively by Streptococcus pneumoniae, whereas CAP is a clinical syndrome with multiple potential etiologies, among which pneumococcus is a predominant cause.
- Line 63 64: and this is throughout the paper when referencing other papers – is to give n/N to help in the interpretation of the % coted.
Response: We have revised the sentence on Page 2 lines 64–65 as follows: “According to the literature, Streptococcus pneumoniae was detected in 29.9% (8,000/26,757) of patients with bacterial infections, and 38.5% (4,049/10,517) among children.” (Page 2, Lines 64–65)
- Line 83: write out CDC
Response: We have written out “CDC” in full as “Centers for Disease Control and Prevention.” (Page 3, Line 102)
- Line 93, and then in methods 102 to 119: please tell us how the stratified sampling was done: random sampling at the population level, or convenience sampling – just give us all the details to help understand what the results may mean.
Response: We apologize for the lack of clarity in our initial description of the data background, which may have caused confusion. These samples were not obtained through population-level random sampling by area. Instead, they are derived exclusively from clinical trials designed to assess the immunogenicity and safety of PCV13, consequently, the serum samples reflect the cohorts enrolled in these specific trials. These trials spanned from 2016 to 2023 and encompassed diverse regions across mainland China, potentially enhancing the representativeness of the study cohort. Additionally, in the methods section, we provided supplementary information on the source of the samples. (Page 3, Lines 121-126)
- Also important is to place the specimens meaningfully in time – so over the 7-year period – how many specimens from which regions were taken in which years. A 7-year period is a long time – and although possible to analyse in a cross-sectional way – you need to convince us of no known pneumococcal outbreaks in the regions from which the individuals came and that vaccination strategies did not change.
- Line 197/198: immediate refer to figure 1 to help the reader.
Response: We have appropriately included Figure 1 in the text to facilitate readers' clear understanding of the regional distribution of the Chinese population and the area covered by our sample collection. (Page 5, Line 225; Page 7, Lines 241-244)
- Line 221 – what is the measure of the statistical significance – a quantitative measure is needed, confidence intervals and then a result for the statistical test etc etc.
- Table 2: it would be good to know the total population for each province, then the number of subjects – just to see if relative proportions of each province. I am not sure if the years can be added here. See where it fits best.
Response: We fully agree that adding the total population of each province in Table 2 helps clarify the relationship between the number of subjects and the local population, which is important for assessing sample representativeness. As suggested, we have added the total population data for each sampling region to Table 2. This allows readers to compare the number of subjects with the corresponding provincial population and better understand the proportional distribution across regions. (Page 6, Lines 237-239)
- Figure 2 C and D are not meaningful?? By stacking the columns there is no data by serotype (the data are not addidative!!) – please redo these figures to show the data meaningfully. So Line 224 to 228 will need to be redone once looking at data more carefully (it contradicts figure 3!). Also define the positive rate is the cut off of >0.35 – so the figures can stand alone.
Response: The stacked column format in Figure 2C and 2D was initially used to show overall trends in antibody levels across regions and age groups, rather than serotype-specific data, which are presented separately in Figure 3.
For example, through the area of color blocks in the stacked chart C below, readers can intuitively and clearly see that the antibody levels against serotype 14 are relatively high across all regions of the country. Meanwhile, the stacked chart D below can directly illustrate that the baseline antibody levels in the population show an obvious U-shaped distribution across different age groups nationwide.
In Figure 2, we have added a title and annotations. The positive rate is defined as the antibody level being greater than 0.35 micrograms per milliliter. (Page 8, Lines 253)
- Figure 2E: what does the grey colour mean – it is not in the key?
Response: Figure 2E. The grey color represents regions where age group data were not available. We apologize for the oversight in the legend. As suggested, we have added a clear explanation of the grey areas in the figure title to ensure that the figure is self-explanatory. (Page 8, Lines 257-260)
- Figure 4 is too small. And needs a careful title, and check how it differs from figure 5. It would be better to choose how best to show difference by age, but not do it in multiple ways. Also see Figure 6. These figures are repetitive and do not help the reader. Rather do two good figures, visible and large enough and carefully interpret the result.
Response: When we were writing the manuscript, we also noticed that Figure 4 was too small. However, after much consideration, we decided to keep this figure and Enlarge it,because:
- China has a vast territory. We need to analyze and compare the differences in the baseline antibody levels among people of the same age group in different regions.
- We did not eliminate the parts that did not show significant differences. The purpose was to enable readers to have a clearer understanding through comparative analysis, especially for vaccine research and development enterprises, as it is of great importance for them to select the clinical collection areas.
To address the issues of small images and large amounts of data, we have made the following improvements:
- We uploaded the original images with higher clarity to ensure they are still clear.
- Added titles to the small pictures, enabling readers to better understand the content depicted in the pictures.
- All the original images have been placed in the supplementary files.
- Figure 7 is again by gender – so links to figure 2 and then work out if differences by gender, and whether truly significant – if not, no need to analyse in such detail. Also see figure 8.
Response: Thank you for your insightful comments on Figures 7 & 8, We have moved Figures 7 and 8 to the supplementary file. Regarding the description of type comparison, we have integrated this part into Figure 2 of the main text for a general description. (Pages 9, Lines 268-279)
- The reader is being flooded by figures – when many fewer would tell the same story – some of the additional analyses could go into supplementary materials.
Response: Thank you for your valuable suggestions. This research study involves over 270,000 pieces of data, which is an extremely large amount. Some parts of the data presented in the manuscript are redundant. We have streamlined and integrated Figure 4, and placed Figures 7 and 8 in the supplementary materials. While retaining the key data as much as possible, some data have been streamlined.
- Discussion: line 331 – 340 – baseline antibody data are useful but not “crucial” for assessing vaccine immunogenicity, and also do not reflect disease – so may not be so helpful in vaccine design and efficacy evaluations – this paragraph would benefit from a more careful discussion – also the whole 1st paragraph says nothing about what the study showed, but are rather sweeping statements with no references.
Response: Regarding the discussion on page 15, lines 380 to 392, we have realized that the previous description failed to fully recognize the significance of the baseline antibody data. We have revised this section and Added a reference to better present the role of the baseline antibody levels to the readers. Although the baseline IgG levels do not directly reflect the disease burden or predict the vaccine efficacy, they still have important uses in certain situations:
- In clinical trial design, they help assess the geographical representativeness of the study population and reduce biases caused by unobserved regional differences.
- Understanding the baseline antibody levels can guide targeted vaccination prevention measures, which is helpful for more effectively utilizing limited resources, especially in cases where vaccine supply or funds are limited.
- The discussion would benefit from paragraphs in themes – comparing the study results to international findings with references.A limitation would be the 7-year period – as much happens in 7 years.
Response: Regarding the potential limitation of the 7-year study period, we understand the reviewer’s concern that various changes may occur over such a long timeframe, including shifts in vaccination strategies and population immunity. The extended duration of this study mainly reflects the integration of experimental data from six vaccine manufacturers. Typically, it takes 3–4 years or more for a single vaccine to complete Phase I–IV clinical trials after approval. The combined timelines from multiple manufacturers naturally prolonged the overall study period.
Importantly, the data were collected in a continuous and standardized manner, ensuring consistency in methods and population characteristics across the study period. Furthermore, based on currently available literature, there have been no large-scale, multicenter baseline surveys on pneumococcal antibody levels over the past seven years, either in China or globally. Therefore, our study helps fill this critical data gap.
We have revised the discussion to include this time span as a limitation, while also explaining the context and continuity of data collection, to present a balanced view of the study’s strengths and limitations.
- Line 391 – why bring in immunogenicity (vaccine response?? Throughout the paper differentiate when discussing antibody response to natural infection vs vaccination, also define carefully when referencing the literature) against serotype 3 – as this was not at all studied in this paper – rather discuss what we know about responses to infection with serotype 3. Your discussion and comparison to the literature need to look at natural infection responses.
Response: We recognize that referencing “immunogenicity” (i.e., vaccine-induced responses) for serotype 3 in line 391 is inappropriate, as our study focused solely on naturally acquired antibodies and did not assess vaccine responses. We have revised the text to ensure accurate use of terminology. (Page 16, Lines 440–445)
Reviewer 2 Report
Comments and Suggestions for Authors
It’s an interesting paper, with a significant number of subjects.
The last paragraph of the Introduction is too large and contains un-needed information. I would keep only the first phrase (lines 89-91).
A paragraph containing information regarding the actual variants of available anti-pneumococcal vaccines and their subtype coverage would be useful in this section.
Lines 221-223: the large sample size makes the statistical data more valid not vice-versa! You should re-phrase
Figure 3 – could the 4 charts be merged in one? Bars and lines with different color according to the region ? The differences would be more easily seen.
Figure 4 – only the significant differences should be displayed
Some of the figures (7, 8?) could go to a supplementary file, not the main article.
Although the number of included children seem high, a reference to the total number of children living in these regions could offer more details on the representativeness of the study.
I would lose the 400-402 lines from the conclusion. Maybe move them in the discussion section.
In the Abstract, the Conclusions are too vague. You could use some of the data from the main article Conclusion section.
Author Response
|
Comments 1: The last paragraph of the Introduction is too large and contains un-needed information. I would keep only the first phrase (lines 89-91). A paragraph containing information regarding the actual variants of available anti-pneumococcal vaccines and their subtype coverage would be useful in this section. |
|
Response 1: Thank you for your insightful comment. We agree that the last paragraph of the Introduction was too lengthy and included non-essential information. In response, we have removed lines 92–95 as suggested. However, we have retained lines 92–97, as this section highlights the significance and rationale of the present study.
|
|
Comments 2: Lines 221-223: the large sample size makes the statistical data more valid not vice-versa! You should re-phrase |
|
Response 2: Thank you very much for your valuable comment. We agree that the original phrasing was ambiguous. We have revised the statement in lines 221–223 to clarify the intended meaning. Specifically, we recognize that from a statistical perspective, when the sample sizes of two groups are large, the power of hypothesis testing increases significantly. This is mainly due to the substantial reduction in the standard error (SE) of the estimated difference between groups, which amplifies the test statistic (e.g., z- or t-value) even when the true effect size is small, thereby leading to a very small p-value. As a result, the null hypothesis (H0) is more likely to be rejected. This highlights a key limitation of over-relying on p-values in large-sample studies. Therefore, we emphasize that effect size and clinical or practical relevance must be jointly considered when interpreting statistically significant findings (Lines 228–233) |
|
Comments 3: Figure 3 – could the 4 charts be merged in one? Bars and lines with different color according to the region? The differences would be more easily seen. |
|
Response 3: Thank you for your valuable suggestion regarding Figure 3. We fully agree that merging the four separate charts into a single figure, using region-specific colors for both bars and lines, would improve clarity and facilitate visual comparison of regional differences. We have revised Figure 3(Lines 260–264) accordingly and updated the manuscript to reflect this change. |
|
Comments 4: Figure 4 – only the significant differences should be displayed Some of the figures (7, 8?) could go to a supplementary file, not the main article. |
|
Response 4: Thank you for your kind suggestions. As most comparisons between groups were statistically significant, we focused on highlighting the non-significant ones to draw attention to where differences were not observed. In addition, each figure presents data for a specific age group, showing detailed results across all serotypes. We also appreciate your advice regarding figure organization. As recommended, we have moved Figure 7 and Figure 8 to the supplementary file to keep the main text more concise. |
|
Comments 5: Although the number of included children seem high, a reference to the total number of children living in these regions could offer more details on the representativeness of the study. |
|
Response 5: Thank you for your suggestion regarding the representativeness of the pediatric sample. We would like to clarify that this is a retrospective study based on serum antibody data from unvaccinated baseline populations enrolled in 13-valent pneumococcal conjugate vaccine (PCV13) clinical trials conducted by six companies. Unlike large-scale epidemiological surveys that cover all age groups or entire populations, our data were strictly limited to the age cohorts specified in the PCV13 trial protocols, in accordance with regulatory requirements for pediatric vaccine evaluation. To enhance transparency regarding sample coverage, we have added information on sampling timeframes and the proportion of enrolled children relative to the total population in each region to Table 2. This addition provides clearer context for interpreting the representativeness of our sample within the framework of clinical trial-based data. We appreciate your attention to this detail, which helped us improve the clarity of our sampling methodology. |
|
Comments 6: I would lose the 400-402 lines from the conclusion. Maybe move them in the discussion section. |
|
Response 6: Thank you very much for your valuable suggestion. We fully agree that the content in lines 400–402 of the conclusion section is more appropriate for the discussion. In the revised manuscript, we have moved them in the discussion section (Lines 356–358). This content was further expanded upon in the discussion section, where we integrated it with relevant research findings and theoretical context to provide a deeper analysis. This adjustment aims to better highlight the study’s implications and limitations, and to enhance the overall logical coherence and depth of the discussion. |
|
Comments 7: In the Abstract, the Conclusions are too vague. You could use some of the data from the main article Conclusion section. |
|
Response 7: We appreciate your constructive suggestion and have revised the abstract’s conclusions section to better reflect the main findings, as shown below: “Baseline IgG levels varied by region and age group. No significant gender differences were observed, and overall antibody levels were higher in the southern region.” This revision incorporates key data from the main conclusion section to enhance clarity and specificity.(Lines 39–40)
|
Reviewer 3 Report
Comments and Suggestions for Authors
Dear authors, you have presented a well-executed large-scale study, yet I have reservations regarding its relevance to developed countries. The PCV13 vaccine is currently regarded as an outdated method of preventing pneumococcal infection in several countries worldwide. Further insights into this provision can be found in the article [Ozisik L. The New Era of Pneumococcal Vaccination in Adults: What Is Next? Vaccines (Basel). 2025;13(5):498. doi: 10.3390/vaccines13050498. PMID: 40432110; PMCID: PMC12115962], as well as in the most recent recommendations issued by the FDA (2024) and ACID (2021). Presently, three PCV13 vaccine products from one imported and two domestic manufacturers are available on the Chinese market, while PCV15, PCV20, PCV24, and PCV26 from more than 10 domestic manufacturers are at different stages of clinical trials (http://www.chinadrugtrials.org.cn/). Furthermore, all three active PCV13 drugs licensed in China are recommended for children aged 6 weeks to 5 years. In light of the aforementioned points, it is recommended that the authors of the article provide a more explicit identification of the target audience for their research, particularly with regard to the challenges posed by public health in China.
Author Response
|
Comments 1: Dear authors, you have presented a well-executed large-scale study, yet I have reservations regarding its relevance to developed countries. The PCV13 vaccine is currently regarded as an outdated method of preventing pneumococcal infection in several countries worldwide. Further insights into this provision can be found in the article [Ozisik L. The New Era of Pneumococcal Vaccination in Adults: What Is Next? Vaccines (Basel). 2025;13(5):498. doi: 10.3390/vaccines13050498. PMID: 40432110; PMCID: PMC12115962], as well as in the most recent recommendations issued by the FDA (2024) and ACID (2021). Presently, three PCV13 vaccine products from one imported and two domestic manufacturers are available on the Chinese market, while PCV15, PCV20, PCV24, and PCV26 from more than 10 domestic manufacturers are at different stages of clinical trials (http://www.chinadrugtrials.org.cn/). Furthermore, all three active PCV13 drugs licensed in China are recommended for children aged 6 weeks to 5 years. In light of the aforementioned points, it is recommended that the authors of the article provide a more explicit identification of the target audience for their research, particularly with regard to the challenges posed by public health in China.
|
|
Response 1: Thank you for your valuable and insightful comments. We fully agree that PCV13 has gradually been replaced by higher-valent pneumococcal conjugate vaccines (e.g., PCV15, PCV20) in several developed countries, as highlighted in the article by Ozisik (2025), and in the latest recommendations issued by the FDA (2024) and ACIP (2021). However, the focus of our study is on healthy children aged 6 weeks to 5 years in China, a population for whom PCV13 remains the only licensed and recommended pneumococcal conjugate vaccine at present. In China, three PCV13 products (one imported and two domestically produced) are currently approved for this age group, and no higher-valent PCVs have yet been authorized for pediatric use. Meanwhile, children under 5 years old in China continue to bear a substantial disease burden from pneumococcal infections. According to estimates, there were approximately 218,200 cases of severe pneumococcal disease and 8,000 related deaths in this age group in 2017, with nearly half of the fatalities occurring in western China. Thus, PCV13 still plays a crucial role in protecting this vulnerable population. Importantly, this study is the first to report baseline IgG antibody levels against all 13 serotypes included in PCV13 in a large, unvaccinated pediatric population across multiple regions in China. Our findings reveal significant regional differences (e.g., higher levels of serotype 14 in southern China) and age-related patterns (e.g., an antibody nadir at 7–11 months), and show that serotypes such as 3 and 4 fall well below the protective threshold of 0.35 μg/mL. These data provide essential scientific evidence to inform the precise and rational application of PCV13 under current national conditions. Although we acknowledge that over ten higher-valent PCV candidates (e.g., PCV15, PCV20, PCV24, PCV26) from domestic manufacturers are in various stages of clinical trials, their immunogenicity and protective efficacy in children remain to be confirmed. Therefore, the baseline immunity data provided in this study are not only relevant to the current use of PCV13 in China but also serve as a reference for future vaccine development and immunization strategy optimization, such as serotype selection and schedule refinement. In summary, this study is clearly targeted at children aged 6 weeks to 5 years in China, and its findings directly address the urgent needs of pneumococcal disease prevention and public health decision-making in this context. |